# Heparanase: A Novel Therapeutic Target for the Treatment of Atherosclerosis

**DOI:** 10.3390/cells11203198

**Published:** 2022-10-12

**Authors:** Tien K. Nguyen, Stephanie Paone, Enoch Chan, Ivan K. H. Poon, Amy A. Baxter, Shane R. Thomas, Mark D. Hulett

**Affiliations:** 1Department of Biochemistry and Chemistry, La Trobe Institute for Molecular Science, La Trobe University, Melbourne, VIC 3086, Australia; 2Department of Pathology, School of Biomedical Sciences, Faculty of Medicine & Health, University of New South Wales, Sydney, NSW 2052, Australia

**Keywords:** heparanase, heparan sulfate, extracellular matrix (ECM), atherosclerosis, inflammation, immune cells

## Abstract

Cardiovascular disease (CVD) is the leading cause of death and disability worldwide, and its management places a huge burden on healthcare systems through hospitalisation and treatment. Atherosclerosis is a chronic inflammatory disease of the arterial wall resulting in the formation of lipid-rich, fibrotic plaques under the subendothelium and is a key contributor to the development of CVD. As such, a detailed understanding of the mechanisms involved in the development of atherosclerosis is urgently required for more effective disease treatment and prevention strategies. Heparanase is the only mammalian enzyme known to cleave heparan sulfate of heparan sulfate proteoglycans, which is a key component of the extracellular matrix and basement membrane. By cleaving heparan sulfate, heparanase contributes to the regulation of numerous physiological and pathological processes such as wound healing, inflammation, tumour angiogenesis, and cell migration. Recent evidence suggests a multifactorial role for heparanase in atherosclerosis by promoting underlying inflammatory processes giving rise to plaque formation, as well as regulating lesion stability. This review provides an up-to-date overview of the role of heparanase in physiological and pathological processes with a focus on the emerging role of the enzyme in atherosclerosis.

## 1. Introduction

Atherosclerosis, a progressive chronic inflammatory disease of the vascular system, is the principal underlying cause of cardiovascular events such as heart attack, stroke, and peripheral vascular disease. As such, atherosclerosis is a major cause of global morbidity and mortality [1]. Heparanase (HPSE) is an endo β-glucuronidase, which is the only mammalian enzyme known to cleave heparan sulfate of heparan sulfate proteoglycans, a key component of vascular extracellular matrix (ECM) and basement membrane. As such, HPSE contributes to various physiological processes associated with heparan sulfate proteoglycan functions and ECM remodelling such as cell–cell interactions, cell migration, wound healing, tissue remodelling, hair growth, and angiogenesis [2,3,4,5,6]. HPSE also plays an important role in several disease settings including cancer metastasis [2,7,8,9,10,11], inflammation [12,13], and diabetes [14,15,16]. Importantly, increasing clinical and experimental evidence highlights a role for HPSE in the development of atherosclerosis [17,18,19]. We presented an overview on the regulatory network of HPSE in health and disease in our previous study [2], which provided a comprehensive review of all known regulators of HPSE expression and activity as well as genes and proteins regulated by HPSE and discussed their relevance in both health and disease. In this review, we summarise the involvement of HPSE in atherosclerosis and discuss possible pathways by which HPSE contributes to its pathogenesis.

## 2. Pathogenesis of Atherosclerosis

Atherosclerosis is a chronic inflammatory disease of large- and mid-sized arteries characterised by the accumulation of cholesterol-rich lipids, fibrous material, and immune cells within the intimal space of arteries, forming complex or advanced atherosclerotic plaques. Advanced, clinically relevant atherosclerotic plaques consist of an acellular necrotic core containing cholesterol crystals and regions of calcification, encapsulated by a fibrous cap of activated and migrated smooth muscle cells (SMCs), monocyte-derived macrophage foam cells, and T cells, that sits below an inflamed layer of vascular endothelial cells (ECs). Unstable plaques can rupture, leading to the formation of an occlusive thrombus in the blood lumen and the resultant precipitation of life-threatening clinical events, heart attack, and stroke [20,21].

Atherosclerosis is a multifactorial disease with complex aetiology and can be divided into three key developmental stages, namely, fatty streak formation (or plaque initiation), plaque progression, and plaque rupture (Figure 1).

### 2.1. Fatty Streak Formation

Endothelial dysfunction is a key event implicated in the initiation of atherosclerosis [22]. The endothelium acts as an interface and functional link between the circulation and the underlying arterial wall. Under normal conditions, a healthy endothelium maintains cardiovascular homeostasis by (i) forming a semipermeable barrier that tonically regulates the transport of nutrients, circulating immune cells and macromolecules such as LDL into the vascular wall, (ii) inhibiting local thrombogenic and inflammatory reactions, (iii) inhibiting SMC migration and proliferation, and (iv) maintaining vascular tone [23,24]. However, in response to pathological atherogenic factors such as hypertension [23], smoking [25], hyperlipidemia [26,27], and disturbed or turbulent blood flow [28,29], the endothelium becomes dysfunctional such that its homeostatic properties are compromised [30,31]. This leads to increased endothelial permeability and infiltration of LDL into the arterial subendothelium and binding to the ECM, enhancing lipoprotein susceptibility to various modifications including oxidation by reactive oxidants generated by infiltrating activated leukocytes [32,33]. Activated ECs also upregulate the expression of proinflammatory leukocyte adhesion molecules and chemokines that signal for endothelial binding and recruitment of circulating monocytes into the intima [33,34].

The migrated monocytes differentiate into macrophages, which can excessively uptake the modified LDL via macrophage scavenger receptors (e.g., SR-A1, SR-A2, CD68, CD36, SR-B1, and LOX-1) [35,36], resulting in the formation of lipid-engorged foam cells, marking the formation of the early fatty streak lesion [33,37]. Macrophages can also process epitopes of internalised modified LDL and present these to CD4^+^ Th-1 cells, thereby stimulating proinflammatory cytokine production. This further activates lesion macrophages and ECs, promoting the production of inflammatory adhesion molecules, cytokines, and chemokines [29,38,39]. These proatherogenic immune processes are counterbalanced by Foxp3^+^ regulatory T cells (Treg cells), which produce anti-inflammatory cytokines such as TGF-β1 and IL-10 [40]. Considerable evidence also supports a complex role for dendritic cells with pro- and anti-atherosclerotic activities for both classical dendritic cells (cDC) [41] and plasmacytoid dendritic cells (pDC) [42,43,44,45] reported, establishing an important role for antigen presentation and adaptive immunity in atherogenesis.

### 2.2. Plaque Progression

With the ongoing persistence of cardiovascular risk factors and Th-1-mediated vascular inflammation, fatty streak lesions are thought to progress into mature or advanced atherosclerotic plaques. Plaque maturation involves the continued formation and accumulation of macrophage foam cells, characterised by dysfunctional lipid metabolism and impaired migratory capacity. These features promote their residence in the intima where they generate various inflammatory agents, namely, cytokines, chemokines, oxidants, and matrix-degrading proteases [46]. While in early lesions, circulating monocytes are recruited into the intimal space where they differentiate to macrophages and take up modified LDL to become foam cells [47]; the increased number of macrophages in more advanced atherosclerotic lesions is not only dependent on monocyte recruitment but also SR-A-initiated proliferation of lesion macrophages [48]. The increase in macrophages or foam cells within an atherosclerotic lesion drives the increase in lesion size and complexity. However, the overall size of a lesion is also determined by the capacity of macrophages to emigrate from atherosclerotic lesions, the degree of foam cell endoplasmic stress and cell death, and the efficiency of lesion macrophages to clear apoptotic or necrotic materials via efferocytosis [46], which is impaired by oxidants and matrix-degrading proteases [49]. Impairment of macrophage efferocytosis leads to the build-up of an acellular necrotic core that is enriched with extracellular lipid and crystalline cholesterol or cholesterol clefts that is released by necrotic foam cells. Formation of the necrotic core is a cardinal feature of advanced or complex atherosclerotic plaques [46].

In parallel with the formation of the necrotic core in advanced or complex plaques, is the formation of a fibrous cap underneath the vascular endothelium, which is composed of ECM proteins such as collagen, elastin, and glycoproteins that are produced by synthetic phenotype vascular SMCs. Synthetic SMCs migrate from the arterial media in response to endothelial- and macrophage-derived promigratory signalling cytokines and growth factors, which stimulate the migration and proliferation of vascular SMCs [50,51,52]. The connective tissue matrix of the fibrous cap acts to provide structural support, and cholesterol, derived from blood lipid or extruded from dying foam cells, becomes entrapped within this matrix [51]. SMC proliferation and migration make an important contribution to the growth and expansion of the atherosclerotic plaque. Notably, SMCs can also differentiate into macrophage-like cells that contribute to lesion foam cell numbers [53]. These processes contribute to the formation of a fibrous cap over the necrotic core, maintaining the continued growth of advanced plaques by isolating the acellular lipid core from the circulating blood [50].

### 2.3. Plaque Rupture and Thrombosis

The formation and growth of atherosclerotic plaques in the arterial wall are driven by complex biological processes. Arterial atherosclerotic plaques are characterised by large and robust fibrous caps that remain stable throughout an individual’s life, which can eventually grow to a size that poses a health risk through stenosis, manifesting clinically as stable angina. However, certain plaques are unstable and, hence, capable of undergoing acute erosion or rupture that precipitate the formation of an occlusive thrombus and the onset of tissue ischemia, the most life-threatening form of which is acute coronary syndromes (ACS) that can manifest as myocardial infarction.

The stability of an atherosclerotic plaque primarily depends upon the size and content of the fibrous cap and necrotic core whereby a low collagen content in the fibrous cap and enlarged, lipid-rich necrotic core favour vulnerability or instability to rupture [54]. The loss of ECM and weakening of the fibrous cap are linked to increased SMC apoptosis and their replacement by macrophages and T cells at the shoulder regions of plaques in parallel with the enhanced release of degradative enzymes such as MMPs by inflammatory macrophages [55,56,57]. Plaque disruption allows exposure of the circulating blood to the highly thrombogenic contents of the necrotic core, promoting platelet aggregation and initiating the formation of a thrombus within the vessel lumen [23], with the potential to obstruct blood flow. Ultimately, plaque rupture is detrimental, leading to ACS and clinical events including heart attack or stroke.

## 3. Current Therapies for Atherosclerosis

Due to the complexity of atherogenesis, limited therapeutic options for disease prevention and treatment exist. Lipid-lowering and blood-thinning therapies, as well as control of cardiovascular risk factors are the most common recommendations of atherosclerotic disease management. Statins are the most widely used cholesterol-lowering drugs, which reduce circulating LDL-cholesterol and triglyceride levels by inhibiting 3-hydroxy-3-methylglutaryl-coenzyme A reductase, which catalyses the rate-limiting step of cholesterol biosynthesis [58]. Statins are generally well-tolerated and lower cholesterol levels more effectively than any available medication. Importantly, in clinical practice, particularly in secondary prevention of atherosclerosis, statins are recommended in all acute myocardial infarction patients without contraindications to statin use, regardless of lipid profile, in order to lower the risk of mortality based on the recommendations of current guidelines [59,60,61,62]. This is due to the pleiotropic effect of statins including stabilisation of the atherosclerotic plaques, anti-inflammatory, antioxidative, and antiproliferative effects [63,64]. However, despite the widespread use of statins for the prevention of atherosclerosis, several challenges remain. For example, the use of statins is associated with muscle symptoms such as muscle pain, aching, cramps, and weakness [65], which are the main reasons why patients discontinue treatment [66]. In addition, some patients taking statins still harbour a discernible residual risk of a cardiovascular event, despite achieving LDL-cholesterol targets [67]. High statin doses are also associated with increased liver transaminase levels and decreased renal function [65]. Furthermore, the use of statins is associated with higher fasting glucose levels and risk of type 2 diabetes mellitus incidence [68].

Proprotein convertase subtilisin/kexin type 9 (PCSK9) inhibitors, a class of lipid-lowering drugs, have been shown to lower LDL-cholesterol effectively [69]. PCSK9 is a serine protease predominantly produced in the liver. Its binding to the LDL receptor (LDL-R) on the surface of hepatocytes triggers LDL-R degradation in lysosomes, leading to higher plasma LDL-cholesterol levels [70]. Therefore, targeting PCSK9 provides an additional pathway to control plasma LDL-cholesterol levels. It has been reported in two phase 3 clinical trials that measured efficacy and safety that two monoclonal antibodies inhibiting PCSK9, evolocumab and alirocumab, significantly lowered LDL-cholesterol by approximately 60% and reduced the incidence of cardiovascular events [68,71,72]. These monoclonal antibodies have also been approved by US Food and Drug Administration and European Medicines Agency for the treatment of elevated plasma LDL-cholesterol [68]. However, their use in clinical practice has been limited due to high costs and the lack of outcomes in large, randomised controlled trials.

In addition to lipid-lowering therapy, blood-thinning treatment including anticoagulant drugs (e.g., heparin and warfarin) and antiplatelet drugs (e.g., aspirin, clopidogrel, prasugrel, and ticagrelor) have also been used for the treatment of atherosclerosis. They are the primary therapies used to prevent atherothrombotic events in CVD patients, particularly in patients who are at the highest risk of cardiovascular events such as myocardial infarction and stroke [73,74]. However, as with statin therapy, the use of blood thinning agents has also been associated with adverse events. Despite continued antiplatelet therapy, recurrent cardiovascular events are still observed in patients, particularly in those at highest risk such as diabetes mellitus and complex coronary artery disease (CAD) [74]. Using anticoagulant agents also increases the risk of dangerous bleeding. Moreover, antiplatelet treatment alone does not reverse remodelling of the atherosclerotic lesions once these lesions are already present [75].

Therefore, alternative therapeutics are needed that can either be taken alone or in combination with the current standard of care for improved treatment for atherosclerosis and its complications.

Recent major clinical trials report that targeting inflammation represents a viable treatment strategy for reducing clinical cardiovascular event risk. For example, the CANTOS and COLCOT trials recently reported that treatment of high-risk CAD patients with a neutralising antibody to IL-1b (Canakinumab) or anti-inflammatory drug colchicine reduced the incidence of recurrent cardiovascular events [76,77]. However, both studies reported an increased risk of life-threatening infections, advocating for more selective targeting of adverse inflammatory events driving atherosclerosis while preserving antimicrobial immunity.

Increasing evidence implicates the enzyme HPSE in different stages of atherosclerosis [19,78,79,80,81]. Importantly, HPSE was suggested to promote inflammation, central to the development of atherosclerosis, by regulating several key processes including immune cell activation, adhesion, and migration [18,82,83,84,85]. Accordingly, the development of HPSE-targeting therapeutics could represent a novel approach for the treatment or prevention of atherosclerosis.

## 4. HPSE and Atherosclerosis

### 4.1. Heparan Sulfate Proteoglycans and HPSE

Heparan sulfate proteoglycans (HSPGs) formed by covalent linkage of a core protein to one or more heparan sulfate (HS) polysaccharides side chains are a major component of ECM and the basement membrane (BM) of all mammalian cells [86]. Within the ECM, HSPGs are involved in a wide range of functional roles in cell physiology including the maintenance of ECM structural integrity, tissue organisation, and BM barrier function via interactions with other ECM components such as proteoglycans, collagen, elastin, and hyaluronic acid [87,88,89]. HSPGs can act as a storage depot for a plethora of bioactive compounds such as growth factors, chemokines, cytokines, enzymes, and inhibitors by virtue of their negatively charged sulfate and carboxylate groups of HS chains [90,91]. The binding of these bioactive molecules to the cell surface and ECM thereby functions in the control of physiological and pathological processes such as morphogenesis, tissue repair, angiogenesis, cancer metastasis, inflammation, atherosclerosis, thrombosis, and diabetes [92,93]. Binding to HS also aids presentation of such molecules on the cell surface, protecting them from proteolysis [94]. Upon cleavage of HS chains, these proteins are liberated and converted into bioactive mediators, ensuring a rapid response to local or systemic cues [20,95]. Additionally, HSPGs mediate cell migration through cell surface stimulatory growth factors bound to HS [95,96] which establishes chemical gradients to facilitate directional recruitment of immune cells [97,98]. Membrane HSPGs can also assist in cell adhesion to ECM, cell–cell interactions, and cell motility through their cooperation with integrins and other cell adhesion receptors, which subsequently contribute to inflammatory responses [87,99]. They can also act as coreceptors for different growth factor receptors, facilitating signal transduction [87,100]. Due to the importance of HSPGs in cellular physiology, uncontrolled degradation of HSPGs could affect important cell signalling pathways and lead to the development of pathological conditions. Importantly, HPSE is the only mammalian enzyme known to directly cleave the HS side chain of HSPGs and, therefore, plays a key role in regulating the release of HSPGs in their various functions.

HPSE is an endo-β-glucuronidase, a member of the glucuronidase family with the ability to degrade HS side chains into 4–7 kDa oligosaccharide fragments that remain biologically active [101]. There are two mammalian HPSE family members: HPSE 1 and 2. Whilst HPSE 1 (referred to as HPSE) can degrade HS, HPSE 2 that shares 35% homology with HPSE 1 has no enzymatic activity [11,102]. The cloning of the HPSE gene was first reported in 1999 [103]. The gene encoding HPSE contains 14 exons and 13 introns and is located on human chromosome 4q22, giving rise to two mRNA species with the same open reading frame, HPSE 1a (1.5 kb) and HPSE 1b (1.7 kb) [104]. HPSE is synthesised as a pre-proenzyme comprising four domains including a signal peptide sequence (Met^1^–Ala^35^), an N-terminal 8 kDa region (Gln^36^–Glu^109^), a 6 kDa linker segment (Ser^110^–Gln^157^), and a hydrophobic C-terminal 50 kDa peptide (Lys^158^–Ile^543^) [103,105]. Removal of the signal peptide results in the secretion of a 65 kDa, inactive pro-HPSE enzyme. After being secreted into the extracellular space, pro-HPSE rapidly binds to cell surface HSPGs, LDL receptor-related protein, or mannose 6-phosphate receptor [106] and is internalised in endosomes, which are converted to lysosomes [107]. In the acidic environment of the lysosome [108], proteolytic removal of the 6 kDa linker segment by cathepsin-L releases an N-terminal 8 kDa (Gln^36^–Glu^109^) and a C-terminal 50 kDa polypeptide (Lys^159^–Ile^543^) [109], with the noncovalent association between these two polypeptide subunits forming the active HPSE heterodimer [105].

Based on sequence identity to specific glycosyl hydrolases, early modelling studies suggested that HPSE adopted a TIM-barrel domain structure composed of eight alpha (α) helices and eight beta (β) sheets [110]. This prediction has since been confirmed by X-ray crystallography, revealing that the structure of human HPSE indeed comprises a (β/α)^8^ TIM-barrel fold [111]. It was also found that the (β/α)^8^ domain is flanked by a smaller β-sandwich domain. Both subunits of the active heterodimer contribute structurally to each domain, with the 8 kDa subunit contributing one β-sheet to the β-sandwich and the first β-α-β fold of the (β/α)^8^ domain, while the 50 kDa polypeptide contributes the remaining folds. Structural determination also confirmed that the HS-binding cleft contains the previously defined residues Glu^225^ and Glu^343^, in addition to basic side chain residues Arg^35^, Lys^158^, Lys^159^, Lys^161^, Lys^231^, Arg^272^, Arg^273^, and Arg^303^ [111]. This detailed understanding of the HPSE structure will facilitate the development of novel therapeutics that target HPSE.

Under physiological conditions, HPSE is expressed at low levels throughout the body. It is preponderantly expressed in placenta and blood-associated cells such as mast cells, neutrophils, and lymphocytes. In disease states, however, elevated HPSE expression is observed, most notably in cancer where it is involved in multiple important roles including extracellular remodelling, tumour angiogenesis, and metastasis [7,112,113,114,115,116,117]. Elevated HPSE expression is also observed in immune cells, such as activated natural killer (NK) cells [118], facilitating the development of several inflammatory diseases such as rheumatoid arthritis [119], inflammatory lung disease [120], and chronic colitis [121].

### 4.2. Physiological and Pathological Functions of HPSE

With the ability to degrade HS, HPSE plays an important role in fundamental biological processes associated with ECM remodelling and cell migration. These include physiological processes such as wound healing, immune surveillance, tissue remodelling, hair growth [3,4,118], exosome secretion [122,123], and autophagy [124], as well as pathological processes in disease development such as cancer metastasis, angiogenesis [11,113], and chronic inflammation [121,125]. The involvement of HPSE in physiological processes and disease development are summarised in the Table 1 and Table 2.

### 4.3. HPSE Expression and Function in Atherosclerosis

HPSE is involved in various physiological and pathological processes, as described above. The involvement of HPSE in tumour progression and metastasis has been extensively studied, however the role of HPSE in CVD, particularly atherosclerosis, is only now emerging. HPSE has been implicated as a mediator in the formation and progression of atherosclerotic plaques as well as the development of vulnerable atherosclerotic lesions. In transgenic mice overexpressing HPSE, fatty streak deposition and arterial thickness are increased [78,79]. Within atherosclerotic lesions, HPSE is expressed in ECs, macrophages, and SMCs [57,190]. Notably, HPSE is overexpressed in vulnerable coronary plaques compared to stable plaques and control nondiseased arteries [81]. Additionally, increased HPSE expression and activity have been reported in human carotid arteries with atherosclerosis [19], in human macrophages treated with ox-LDL and angiotensin II [78], or ECs treated with inflammatory cytokines [190]. Since atherosclerosis is a chronic inflammatory disease of the arterial system, it is important to address the effect of HPSE on inflammation as well as the involvement of HPSE in the pathogenesis of atherosclerosis in a vasculature cell-type specific manner. The potential roles of HPSE in the pathogenesis of atherosclerosis are depicted in Figure 2.

#### 4.3.1. HPSE Modulates Vascular Inflammation by Modifying Immune Cell Activation, Migration, and Adhesion

Inflammation is an immune response that is triggered by noxious stimuli and conditions such as infection and tissue injury [191]. Previous studies have demonstrated the key involvement of HPSE in inflammation. In the presence of HPSE, a variety of proinflammatory cytokines in both human and mouse immune cells are upregulated in vivo [81,171]. Likewise, HPSE expression is induced in the presence of inflammatory cytokines. Key proinflammatory cytokines such as IFN-γ, IL-1β, and TNF-α induce HPSE mRNA expression and activity in vascular ECs via a caspase 8-dependent pathway [190]. Furthermore, the anti-inflammatory effects of HPSE inhibitors such as heparin and synthetic heparin-mimicking compounds in clinical and animal studies further support the involvement of this enzyme in inflammation [184,192].

Immune cell activation, recruitment, and adhesion to sites of injury are key contributing factors in the inflammatory response. Not only do mature immune cells need to migrate towards the area of insult, adhesion to the endothelium is also required for extravasation into the tissue. Although the activation, recruitment, and adhesion of immune cells are influenced by many factors, HPSE has also been shown to contribute to these processes in various ways. The initiation of adaptive immunity begins when DCs become activated and confer the ability to present foreign antigens to T lymphocytes. Exogenous agonists such as lipopolysaccharide (LPS) [193] or proinflammatory cytokines such as TNFα [194] and activated leukocyte-derived IL-1 [195] can activate DCs. Interestingly, HS fragments alone are also able to activate DCs [127]. Likewise, when incubated with HS fragments generated by HPSE, mononuclear cells release proinflammatory cytokines [128]. In addition to HS fragments, HPSE itself can also directly mediate immune cell activation. In macrophages, the binding of HPSE to TLR receptors has been suggested to initiate cellular activation along with the induction of proinflammatory mediators such as TNFα, IL-1, and MCP-1 [81]. HPSE was also found to promote macrophage activation and cytokine expression via Erk, p38, and JNK signalling [18]. Therefore, through either indirect (HS cleavage) or direct (receptor binding) mechanisms, HPSE appears to contribute to immune cell activation associated with upregulation of proinflammatory cytokines, playing an important role in the inflammatory process.

Inflammation is mediated by the migration of leukocytes to the affected area and relies on the release of chemokines and cytokines by cells at inflammatory sites to facilitate their recruitment. HPSE has been implicated in these processes due its ability to break down the ECM, either by being secreted [151] or maintained on the cell surface [149]. This allows immune cells to move through the vasculature into the area of inflammation, which in the context of atherosclerosis is the arterial intima or subendothelial layer. HPSE also supports the recruitment of leukocytes to inflammatory sites by releasing HS-bound growth factors and cytokines within the ECM or cell surfaces [196,197]. In addition to leukocyte recruitment, HPSE can induce EC migration via protein kinase B/Akt activation [84]. HPSE also plays an important nonenzymatic, promigratory role in macrophage recruitment during inflammation through the inhibition of AGE-stimulated macrophage migration-specific inhibition of the nonenzymatic terminal of HPSE [198]. In vivo studies also support the role of HPSE in immune cell migration, demonstrating a significant inflammatory cell infiltration into the peritoneal cavity of rats upon intraperitoneal administration of recombinant HPSE [85]. Furthermore, HPSE-deficient mice display reduced DC [150] and eosinophil [137] recruitment to the airways in mouse models of pulmonary inflammation.

The migration of immune cells to inflammatory sites is a tightly regulated process that requires cells to adhere to the endothelial surface of the blood-vessel wall before migrating through the BM. The involvement of HPSE in cell adhesion has also been demonstrated in several previous studies. ECM-bound inactive HPSE captures T cells through interaction with T cell-surface HSPGs, mediating rolling and arrest in a α4β1-vascular cell adhesion molecule-1 dependent manner [131]. Additionally, the two functional heparin-binding domains of HPSE (Lys^158^-Asp^171^ and Gln^270^-Lys^280^) mediate syndecan clustering on migrating cells that stimulate the adhesion and spreading of multiple cell types by activation of the Rac1, Src, and PKC pathways [83]. In the presence of HPSE, adhesion of neutrophils and mononuclear cells to ECs in vitro is also enhanced [85]. Through the HS-degrading ability, HPSE enhances leukocyte access to adhesion molecules expressed within the EC surface, contributing to neutrophil adhesion to the pulmonary endothelium during lung injury [120]. A role of HPSE in immune cell adhesion is further supported by the finding that HPSE inhibitor heparin can prevent the adhesion of mononuclear cells to ECs [199].

#### 4.3.2. HPSE Alters the Homeostasis of ECs

Although atherosclerosis is a complex, multifactorial process, changes to endothelial homeostasis are considered to precede disease development [200,201]. Through its degradative capacity, it is suggested that HPSE mediates changes to the permeability of the endothelium, allowing the storage of LDL in the vessel wall, which is a key initiating factor in the formation and development of atherosclerosis [162]. Storage of LDL in the subendothelial layer not only mediates the recruitment of monocytes, but its oxidisation and ingestion by macrophages also results in the formation of foam cells [202], which are key events to plaque initiation and development. LDL retained in the intima also activates the endothelium, increasing HPSE expression [162]. Subsequent degradation of HSPGs compromises the barrier function of the endothelium and facilitates additional lipid binding [162], initiating a proinflammatory, proatherosclerotic cycle. In addition to LDL, high glucose [27] and proinflammatory cytokines TNFα and IL-1β [190] also induce HPSE expression by ECs, thereby further increasing the degradation of HSPGs [129] and promoting the adhesion of monocytes in vitro [199]. Furthermore, endothelial adhesion of peripheral blood mononuclear cells (PBMCs) is prevented in the presence of heparin, an HPSE inhibitor [199], suggesting that damaged endothelium releases HPSE and this released HPSE facilitates monocyte recruitment and binding.

Another important determinant of atherosclerosis affecting the endothelium is low shear stress (LSS) [203]. Shear stress mediates various biological responses by the endothelium and is detected by cell-surface HSPGs [204]. Regions of the vasculature subjected to laminar flow or high shear stress (HSS) are athero-resistant, exhibiting decreased plaque development [205]. It has been suggested that HSS activates signal transduction leading to expression of anti-atherosclerotic factors [206]. In regions of HSS, an increased thickness of the HSPG-rich glycocalyx has been observed [207]. These observations suggest that in regions of the vasculature under HSS, degradation by proteases such as HPSE is limited, preventing atherosclerotic plaque development. Conversely, in regions of vessel branching, bifurcations, and curvatures, blood flow is turbulent and gives rise to oscillatory flow, or LSS [205]. Regions of the vasculature subjected to LSS are more susceptible to the development of atherosclerotic plaques and, interestingly, elevated HPSE expression is limited to these athero-prone regions [57]. It has also been demonstrated that in areas of LSS, levels of the HSPG syndecan, which is required for EC alignment and maintenance of barrier integrity, are reduced [208]. HSPGs within the glycocalyx are also linked with the control flow-mediated production of endothelial-derived nitric oxide (NO), which is critical for maintenance of endothelial and cardiovascular homeostasis [209]. Therefore, HSPG reduction due to LSS-mediated HPSE expression could, therefore, compromise the function of the endothelium, leading to atherosclerosis.

#### 4.3.3. HPSE Promotes Monocyte Binding- and Macrophage-Mediated Inflammation

In addition to ECs, monocytes and macrophages are also pivotal to the development of atherosclerosis [210]. Monocyte recruitment from the blood stream and adhesion to the activated endothelium are crucial steps in the initiation of atherosclerosis [211]. As mentioned previously, HPSE can mediate monocyte binding to ECs. Interestingly, increased binding is observed following the treatment of ECs with atherogenic lipids such as lysolecithin [129], suggesting that ECs may secrete HPSE in response to dyslipidaemia to promote HSPG degradation [120]. The binding of monocytes to the endothelium can be prevented upon treatment with heparin [199], further supporting the involvement of HPSE in facilitating binding and subsequent retention of monocytes in the vessel wall during atherosclerosis. Once retained in the vessel wall, differentiation of monocytes into macrophages occurs, with HPSE also suggested to contribute to this process [18]. Depending on the stimulus, differentiation can result in the generation of two distinct macrophage subtypes, the proinflammatory M1 or the alternatively activated M2 [212]. Within the atherosclerotic plaque, release of cytokines and oxidants along with the generation of MMPs are characteristic of M1 macrophages, contributing to the proinflammatory nature of the plaque [213]. Importantly, in vitro treatment of macrophages with purified HPSE causes cellular activation and upregulation of proinflammatory cytokines and chemokines such as TNFα, IL-1, and MCP-1, as well as proteases such as MMP-9 [81], suggesting that HPSE-mediated activation of macrophages can promote the establishment of inflammation by polarizing macrophages towards the proinflammatory M1 phenotype.

#### 4.3.4. HPSE Promotes SMC Proliferation and Plaque Rupture

SMC proliferation and migration are important events in the progression of fatty streaks to fibrous plaques. Mature SMCs are not terminally differentiated and, thus, retain the ability for rapid phenotypic change into the synthetic form that migrates and releases large amounts of ECM [52]. In a rabbit model of EC injury, MMPs were suggested to act in concert with HPSE to promote phenotypic switching of SMCs, ultimately contributing to myointimal thickening and progression to complex atherosclerotic lesions [163]. In a similar model of EC injury, inhibition of HPSE with the sulfated oligosaccharide PI-88 limited SMC migration via binding to FGF-2 [181]. Although SMC proliferation and migration mediates the formation of a fibrous cap that serves to contain and stabilise the growing plaque, over time, plaque rupture can occur because of degradative enzymes [55]. HPSE staining of unstable carotid atherosclerotic plaques colocalised with other proteases such as MMPs and PCSK9 suggesting that the enzyme contributes to degradation of the SMC fibrous cap [214], playing a key role in atherosclerotic plaque rupture. Interestingly, coronary specimens obtained from vulnerable plaques show significantly increased HPSE staining compared with specimens of stable plaques and healthy arteries. In addition, plasma HPSE levels are much higher in patients with acute myocardial infarction compared to patients with stable angina and heathy individuals [81].

Collectively, this clinical evidence indicates a strong association of HPSE with atherosclerotic plaque progression and instability, and it may, thus, be considered as a diagnostic marker and potentially therapeutic target in atherosclerotic disease.

## 5. HPSE in Models of Atherosclerosis and Therapeutics Targeting HPSE

The apolipoprotein E knockout (ApoE^−/−^) mouse is currently the most widely used model for the study of human atherosclerotic disease [215]. ApoE is a key component of high-density lipoprotein (HDL) and regulates plasma cholesterol by facilitating the uptake and clearance of lipoproteins [216]. Accordingly, ApoE-deficient mice exhibit five times the normal plasma cholesterol and triglyceride levels and develop spontaneous atherosclerotic lesions [217,218].

HPSE overexpressing transgenic mice on an ApoE^−/−^ background have been created to investigate the role of HPSE in atherosclerotic disease development in vivo. Atherosclerotic lesion development is reportedly accelerated in transgenic mice and such lesions have reduced stability due to exacerbated collagen degradation [219]. Furthermore, accelerated disease development was also observed in ApoE^−/−^ mice transplanted with bone marrow from HPSE transgenic mice, highlighting a proatherogenic role for HPSE contributed by cells of the hematopoietic system [219]. HPSE overexpression in mice has also been linked to reduced hepatic clearance of triglyceride-rich particles (TRPs) in association with increased fatty streak area [79]. It was suggested that HPSE degradation of HS limits the hepatic binding and clearance of TRP, demonstrating the importance of HS in triglyceride uptake for the prevention of early atherosclerotic lesion development.

While animal models have been used to mimic human disease and the results obtained from these models highlight the role of HPSE in atherosclerosis, the clinical relevance of HPSE is indispensable when reviewing human studies. Plasma levels of HPSE are elevated in myocardial infarction patients compared to those with stable angina or healthy controls [81]. Additionally, increased HPSE staining is observed in vulnerable plaques compared to stable plaques [81] where HPSE is found in association with foam cells within the vessel wall [220]. Increased HPSE has been observed in carotid atherosclerotic lesions compared to unaffected vasculature, as well as in atherosclerotic lesions from symptomatic patients compared to asymptomatic individuals [19]. Although an exact mechanism of action for HPSE in atherosclerosis has not been elucidated, preliminary results from mouse models and evidence of HPSE present in human atherosclerotic samples strongly suggest that HPSE contributes to atherosclerotic disease development. The above studies advocate for more detailed studies into the atherogenic activity of HPSE and its mechanisms of action.

In addition to in vivo HPSE overexpression studies, HPSE inhibitors have also been used in animal models of atherosclerosis. An HS-mimetic HPSE inhibitor, PG545, shows high potency in preclinical cancer models and is currently being assessed in clinical trials in patients with advanced solid tumours [221]. Recent studies have used PG545 for the treatment of atherosclerosis in high fat-fed ApoE^−/−^ mice and showed promising outcomes including reduced blood pressure, blood glucose, serum oxidative stress [222], total cholesterol, LDL, and triglycerides along with a reduction in aortic wall thickness, atherosclerosis development, and liver steatosis [17] despite the existence of some potential detrimental effects such as reduced HDL serum level [222] or attenuated weight gain [17]. Further studies are, however, required to verify the specificity of PG545 for HPSE. However, these studies once again highlight the potential roles of HPSE in atherosclerosis and the utility of HPSE inhibitors could be an effective strategy for the treatment of CVD.

## 6. Concluding Remarks and Future Directions

Despite current treatments, atherosclerosis remains a major risk factor for CVD, the leading cause of death globally, prompting an ongoing urgent need for novel therapies. Since the cloning of human HPSE in 1999, the enzyme was shown to be involved in a vast range of physiological and pathological processes. While most attention has been placed on the role of HPSE in tumour progression and metastasis, emerging evidence indicates that HPSE also contributes to several inflammatory disease conditions including inflammatory bowel disease, rheumatoid arthritis, type 1 diabetes, psoriasis, diabetic nephropathy, and atherosclerosis. As outlined within this review, HPSE is implicated as a contributing factor not only to atherogenesis, but also plaque development from initial stage through to later stage disease presentations, emphasising its important roles throughout atherosclerotic disease progression. These findings, therefore, highlight HPSE as an attractive therapeutic target for this chronic inflammatory disease.

Despite an increasing trend towards the use of experimental animal models as well as studies involving pharmacological inhibition of HPSE, the importance and precise roles of HPSE in controlling atherosclerosis and the mechanistic insights into how HPSE promotes this disease remain largely unknown. Importantly, no study has employed the use of an HPSE genetic ablation mouse model to characterise the development and progression of atherosclerosis. Such a model, involving HPSE gene-deficient mice on the atherosclerosis prone ApoE^−/−^ background would be an essential tool with which to gain a deeper understanding of the precise pathogenic role HPSE plays in driving atherosclerosis. Critically, the knowledge gained through this type of genetic study would support the development of novel HPSE-targeting drugs for preventing atherosclerotic plaque formation in the treatment of CVD.

## Figures and Tables

**Figure 1 cells-11-03198-f001:**
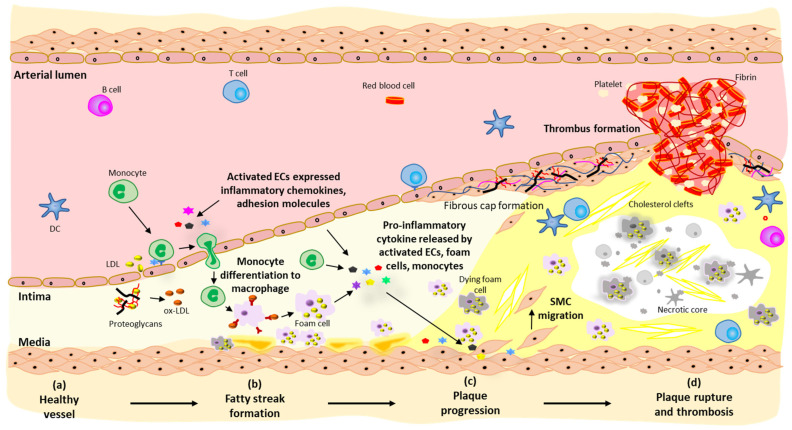
Pathogenesis of atherosclerosis. (**a**) Normal structure of an artery with an intact endothelium separating the arterial wall from the circulating blood. (**b**) Endothelial dysfunction in response to increased cardiovascular risk factors (e.g., hyperlipidemia, hyperglycemia, hypertension) compromises its restrictive barrier function, allowing the enhanced entry of low-density lipoprotein (LDL) into the arterial intima where it can become trapped through binding to the ECM, increasing the susceptibility of LDL to oxidative modification. Activated ECs also express proinflammatory adhesion molecules and chemokines, which facilitate monocyte recruitment into the arterial intima and their differentiation into macrophages. The excessive uptake of lipids from modified LDL results in the formation of foam cells, marking the formation of early fatty streak lesions. (**c**) Recruited monocytes/macrophages and the activated, dysfunctional endothelium release cytokines and growth factors, which stimulate the migration and proliferation of medial vascular SMCs. These ‘synthetic’ SMCs secrete large amounts of ECM (primarily collagen) to form a subendothelial fibrous cap overlying a developing acellular necrotic core that is derived from the release of large amounts of extracellular cholesterol from dying foam cells, constituting an advanced atherosclerotic lesion. (**d**) Continued growth of the necrotic core coupled with ongoing arterial inflammation, accumulation of macrophages and T cells in the plaque shoulder regions, and release of ECM-degrading matrix metalloproteinases (MMPs) leads to thinning and weakening of the collagen-rich fibrous cap, leading to plaque rupture. Exposure of the necrotic core to the circulating blood initiates complement activation and thrombus formation, which can occlude arterial blood flow.

**Figure 2 cells-11-03198-f002:**
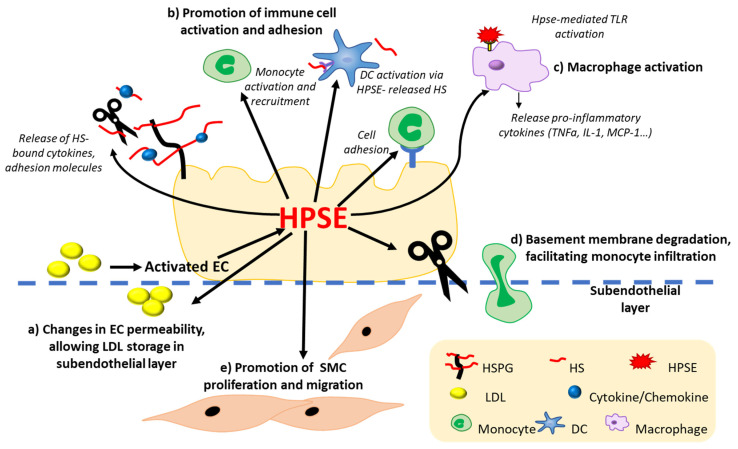
Potential roles of HPSE in atherosclerosis. (**a**) HPSE released from activated ECs mediates changes to the permeability of the endothelium, allowing excessive storage of LDL in the vessel wall. (**b**) By releasing HS fragments, HS-bound cytokines, adhesion molecules, and growth factors, HPSE promotes the activation and adhesion of DCs and monocytes to inflammatory sites. (**c**) HPSE binds to toll-like receptors (TLRs) in the surface of macrophages, facilitating macrophage activation and inflammatory cytokine production. (**d**) HPSE degrades the BM, facilitating binding and infiltration of monocytes to the subendothelial layer. (**e**) EC-released HPSE promotes SMC proliferation and migration, mediating the formation of a fibrous cap.

**Table 1 cells-11-03198-t001:** HPSE functions in physiological processes.

Physiological Process	HPSE Function	References
Angiogenesis	Liberates HS-bound angiogenic growth factors VEGF and basic FGF	[5,126]
Autophagy	Confers resistance to chemotherapy through regulation of autophagy	[124]
Cell activation	Releases soluble HS, HS-bound growth factors, activating DCs and macrophages; signals through toll-like receptors, activating macrophages	[18,81,127,128]
Cell adhesion	Facilitates platelet, neutrophil, monocyte, T cell, and hematopoietic progenitor cell adhesion independent of enzymatic activity	[82,83,120,129,130,131,132,133,134]
Cell proliferation	Stimulates chondrogenesis in bone growth plates	[135]
Cell recruitment	Stimulates eosinophil recruitment to the lungs; influences chemokine gradient established by endothelial HS to alter neutrophil crawling	[85,136,137]
Cell signalling	Contributes to cell signalling via Akt, PIK3, ERK, JNK, P38, Src	[18,84,126,138,139]
Coagulation	Stimulates tissue factor expression by endothelial and cancer cells facilitating coagulation; dissociates tissue factor pathway inhibitor, increasing cell surface coagulation activity; increases generation of activated factor X; activates anti thrombin at physiologic pH	[140,141,142,143]
Exosome biogenesis	Activates syndecan–syntenin–exosome biogenesis pathway and regulates exosome production, composition, and secretion	[122,123]
Hair growth	Increases vascularisation in the hair follicle, migration of follicular stem cell progeny, and release of HS-bound growth factors regulating the hair growth cycle; contributes to hair follicle cycling through HS regulation	[3,144,145]
Lymphangiogenesis	Upregulates VEGF and cyclooxygenase-2	[146,147]
Migration	Mediates migration of DCs, neutrophils, T cells, monocytes, microglia, and ECs	[84,131,148,149,150,151]
Reproduction	Endometrial remodelling essential for embryo implantation; regulates tissue factor and tissue factor pathway inhibitor in trophoblasts during early miscarriages; trophoblast invasion; placental maturation	[152,153,154,155]
Transplant rejection	Inhibits T cell activation and modulates cytokine release to facilitate engraftment and suppress graft-versus-host disease	[156]
Viral release	Cleaves HS-bound viral progenies to facilitate release of HSV-1, HSV-2, HPV, and PRRSV, preventing viral infection	[157,158,159,160]
Wound healing	Increases vascularity to accelerate wound closure	[3,5,161]

**Table 2 cells-11-03198-t002:** HPSE functions in disease development.

Disease	HPSE Function	References
Airway inflammation	Facilitates DC and eosinophil recruitment and allergic inflammatory response; contributes to sepsis-associated acute lung injury by promoting pulmonary glycocalyx loss and neutrophil adhesion	[120,137,150]
Atherosclerosis	Contributes to LDL retention in the intima; facilitates monocyte binding to the endothelium; mediates SMC proliferation and migration; promotes release of inflammatory mediators from macrophages	[81,129,162,163]
Crohn’s/colitis	Stimulates macrophage activation	[121]
Diabetes	Mediates pancreatic β-cell death and macrophage activation; regulates fatty acid use by cardiomyocytes; mediates proteinuria through degradation of the nephron BM	[164,165,166,167,168]
Experimental autoimmune encephalomyelitis	Promotes T cell activation; induces upregulation of Th2 cytokines, inhibits inflammation; facilitates CD4^+^ T cell infiltration into the CNS, promoting inflammation	[169,170,171]
Glomerular disease	Promotes HS degradation, increasing glomerular BM permeability, enhancing leukocyte and macrophage influx; upregulates inflammatory mediators	[172,173,174,175]
Ischemia reperfusion injury	Regulates FGF-2 and transforming growth factor β-induced epithelial to mesenchymal transition, inducing chronic renal damage; contributes to revascularisation by promoting angiogenesis; promotes astrocyte migration towards ischemic core, resulting in astrogliosis; regulates macrophage polarisation to inflammatory M1 phenotype	[176,177,178,179,180]
Neointimal hyperplasia	Promotes SMC proliferation to establish restenosis; stimulates macrophage recruitment and vascular remodelling	[78,181]
Neuroinflammation	Prevents immune cell recruitment, activation, and clearance of amyloid-β; delays prion disease onset and progression; increases leukocyte trafficking associated with neurological deficiency after subarachnoid haemorrhage	[182,183,184]
Psoriasis	Facilitates activation of skin-infiltrating macrophages and induction of signal transducer and activator of transcription 3, enhanced NF-κB signalling, and increased TNFα expression	[185]
Sepsis	Facilitates destruction of mucosal epithelial glycocalyx via HS degradation, promoting neutrophil infiltration and inflammatory cytokine production, acts as a predictor of sepsis severity	[13,186,187]
Sinusitis	Contributes to tissue remodelling in nasal polyps	[188]
Rheumatoid arthritis	Regulates angiogenesis and stimulates immune cell migration and proliferation	[119,189]

## Data Availability

Not applicable.

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
