# Peer review of "Heparanase: A Novel Therapeutic Target for the Treatment of Atherosclerosis"

_cells, 2022, doi:10.3390/cells11203198_

Round 1

Reviewer 1 Report

This is an excellent review article on Heparinase as A Novel Therapeutic Target for the Treatment of  Atherosclerosis

The authors nicely presented pathogenesis of atherosclerosis as text and Figure 1, incuding fatty streak formation, plaque progression, rupture and subsequent thrombosis as well as current therapies of atherosclerosis.

The second part is dedicated to heraranase and its effect upon atherosclerosis. The mechanisms are explained in details. In table 1 the role of heparanae in physiological processes is listed and in table 2 the role in disease states.

The role of heparanase in atherosclerosis is nicely presented in Figure 2 and explained in the text.

The references are appropriate

My comments:

In the section Current therapies of atherosclerosis  the authors listed anticoagulants and antiplatelet therapy. However, in current clinical practice antithrombotic therapy is mainly  started in complications of atherosclerosis such as acute MI, stroke, when thrombosis developes upon ruptured , eroded, or infammed plaque. Before the events we are not aware of the presence of atherosclerosis, but we assume that there is an increased risk for atherosclerosis and its complications in patients with risk factors (hypertension, diabetes, diyslipidemia, obesity, inactivity, older age, etc.).

In addition, in the same section, the authors stress the importance of statins, in particular by lowering holesterol. In current clinical practice statins, in particular in secondary prevention of atherosclerosis, are administered to each patient with AMI, irrespective of the lipid profile due to the pleiotropic effect of statins, which includes stabilization of the atherosclerotic plaques, anti-inflammatory, antioxidative, antiproiferative effect.  This should be mentioned as well

Author Response

This is an excellent review article on Heparanase as A Novel Therapeutic Target for the Treatment of  Atherosclerosis. The authors nicely presented pathogenesis of atherosclerosis as text and Figure 1, including fatty streak formation, plaque progression, rupture and subsequent thrombosis as well as current therapies of atherosclerosis. The second part is dedicated to heparanase and its effect upon atherosclerosis. The mechanisms are explained in details. In table 1 the role of heparanase in physiological processes is listed and in table 2 the role in disease states. The role of heparanase in atherosclerosis is nicely presented in Figure 2 and explained in the text. The references are appropriate.

Comments and suggestions for authors:

In the section Current therapies of atherosclerosis  the authors listed anticoagulants and antiplatelet therapy. However, in current clinical practice antithrombotic therapy is mainly  started in complications of atherosclerosis such as acute MI, stroke, when thrombosis develops upon ruptured , eroded, or inflamed plaque. Before the events we are not aware of the presence of atherosclerosis, but we assume that there is an increased risk for atherosclerosis and its complications in patients with risk factors (hypertension, diabetes, dyslipidaemia, obesity, inactivity, older age, etc.).

Response: Thank you for your helpful suggestion. We agree and have incorporated a section to address this point in the revised manuscript. The changes to the manuscript are highlighted using Track Changes at section: ‘Current therapies of atherosclerosis’, pages 4-5, lines 176-181.

In addition, in the same section, the authors stress the importance of statins, in particular by lowering cholesterol. In current clinical practice statins, in particular in secondary prevention of atherosclerosis, are administered to each patient with AMI, irrespective of the lipid profile due to the pleiotropic effect of statins, which includes stabilization of the atherosclerotic plaques, anti-inflammatory, antioxidative, anti-proliferative effect.  This should be mentioned as well

Response: Again, we thank the reviewer for raising this important point. We have revised our manuscript accordingly and highlighted using Track Changes at section: ‘Current therapies of atherosclerosis’, page 5, lines 205-208.

Reviewer 2 Report

Thank you for the opportunity to review this paper. It is well written, interesting and relevant. There are a few suggestions that I think would improve the quality of this review. Two of the authors on this paper have written a earlier review titled "The Heparanase regulatory network in health and disease" so this review seems to be an extension of the theme of the previous review. This can be mentioned in the paper without self citation so the readers can understand the framework of the current review better. The review could also benefit from adding the review by Khanna (Heparanse: Historical aspects and future perspectives Adv Exp Med Biol. 2020; 1221:71-96. doi: 10.007/978-3-030-34521-1_3) since that provides a good background on heparanase. 

Author Response

Thank you for the opportunity to review this paper. It is well written, interesting and relevant. There are a few suggestions that I think would improve the quality of this review. Two of the authors on this paper have written a earlier review titled "The Heparanase regulatory network in health and disease" so this review seems to be an extension of the theme of the previous review. This can be mentioned in the paper without self-citation so the readers can understand the framework of the current review better. The review could also benefit from adding the review by Khanna (Heparanase: Historical aspects and future perspectives Adv Exp Med Biol. 2020; 1221:71-96. doi: 10.007/978-3-030-34521-1_3) since that provides a good background on heparanase. 

Response: We thank the reviewer for the helpful suggestion. We agree and have revised the manuscript as suggested. A sentence describing our previous review and how it relates to the current review has been added to the ‘Introduction’ section (highlighted in the revised manuscript through Track Changes pages 1-2, lines 42-46). We have also added the review by Khanna as reference [11] (highlighted) in sections: ‘Introduction’ (page 1, line 40), ‘Heparan sulfate proteoglycans and HPSE’ (page 6, line 265) and ‘Physiological and pathological functions of HPSE’ (page 7, line 306).